

# Can downwelling far-infrared radiances over Antarctica be estimated from mid-infrared information?

Christophe Bellisario[1], Helen E. Brindley[2], Simon F.B. Tett[1], Rolando Rizzi[3], Gianluca Di Natale[4], Luca Palchetti[4], and Giovanni Bianchini[4]

[1]School of Geosciences, University of Edinburgh, Crew Building, The King's Buildings, Edinburgh EH9 3FF, UK
[2]Space and Atmospheric Physics Group, National Centre for Earth Observation, Imperial College London, London, UK
[3]Department of Physics and Astronomy, University of Bologna, Bologna, Italy
[4]Istituto Nazionale di Ottica - CNR, Sesto Fiorentino, Italy

**Correspondence:** Christophe Bellisario (christophe.bellisario@ed.ac.uk)

**Abstract.** Far-infrared (FIR: 100 cm$^{-1}$ < wavenumber, $\nu$ < 667 cm$^{-1}$) radiation emitted by the Earth and its atmosphere plays a key role in the Earth's energy budget. However, because of a lack of spectrally resolved measurements, radiation schemes in climate models suffer from a lack of constraint across this spectral range. Exploiting a method developed to estimate upwelling far-infrared radiation from mid-infrared (MIR: 667 cm$^{-1}$ < $\nu$ < 1400 cm$^{-1}$) observations, we explore the possibility of inferring

zenith FIR downwelling radiances in zenith-looking observation geometry, focusing on clear-sky conditions in Antarctica. The methodology selects a MIR predictor wavenumber for each FIR wavenumber based on the maximum correlation seen between the different spectral ranges. Observations from the REFIR-PAD instrument (Radiation Explorer in the Far Infrared - Prototype for Application and Development) and high resolution radiance simulations generated from co-located radio soundings are used to develop and assess the method. We highlight the impact of noise on the correlation between MIR and FIR radiances

by comparing the observational and theoretical cases. Using the observed values in isolation, between 150 and 360 cm$^{-1}$, differences between the 'true' and 'extended' radiances are less than 5 %. However, in spectral bands of low signal, between 360 and 667 cm$^{-1}$, the impact of instrument noise is strong and increases the differences seen. When the extension of the observed spectra is performed using regression coefficients based on noise-free radiative-transfer simulations the results show strong biases, exceeding 100 % where the signal is low. These biases are reduced to just a few percent if the noise in the

observations is accounted for the simulation procedure. Our results imply that while it is feasible to use this type of approach to extend mid infrared spectral measurements to the far-infrared, the quality of the extension will be strongly dependent on the noise characteristics of the observations. A good knowledge of the atmospheric state associated with the measurements is also required in order to build a representative regression model.

## 1  Introduction

Defined here as wavelengths above 15 $\mu$m or wavenumbers below 667 cm$^{-1}$, the far-infrared (FIR) spectral band plays a key role in energetic exchanges between the Earth's surface, atmosphere and space (Harries et al., 2008). Under clear-sky conditions, absorption in the FIR is dominated by water vapour such that typically very little FIR radiation emitted from the surface directly





escapes to space. However, the very cold, dry conditions commonly found in polar regions simultaneously shift the peak of surface emission towards longer wavelengths and, under clear-skies, allow a significant fraction of FIR radiation emitted from the ground to escape directly to space, making the clear-sky FIR outgoing longwave radiation sensitive to surface properties (Feldman et al., 2014). A corollary of this enhanced atmospheric transmissivity is the increased sensitivity of downward clear-

sky FIR radiation at the surface to conditions at higher levels in the atmosphere than would normally be the case in warmer, wetter environments.

Despite its role in the energy budget, due to the inherent difficulties involved, only a few instruments have measured hyperspectral radiances across the FIR. Aircraft and ground based measurements available from the Tropospheric Airborne Fourier Transform Spectrometer (TAFTS) (Canas et al., 1997) vvvhave been used to probe water vapour spectroscopy; upper tropo-

spheric humidity; the radiative properties of cirrus and snow/ice surface emissivity (Green et al., 2012; Fox et al., 2015; Cox et al., 2007; Cox et al., 2010; Bellisario et al., 2017). Balloon and ground-based observations from the Radiation Explorer in the Far InfraRed - Prototype of Applications and Development (REFIR-PAD, Bianchini et al., 2006) have been exploited to determine precipitable water (Bianchini et al., 2011), investigate the spectral signature of cirrus (Maestri et al., 2014) and provide simultaneous retrievals of water vapour, temperature and cirrus properties (Di Natale et al., 2017). Finally, the Far-

InfraRed Spectroscopy of the Troposphere (FIRST) instrument (Mlynczak et al., 2006) has participated in both balloon and ground-based campaigns, providing a rigorous test of the ability of radiative transfer models to match the spectroscopic signals measured in the far infra-red (Mlynczak et al., 2016; Mast et al., 2017).

Almost all of the available FIR radiance measurements originate from limited field campaigns. Recognising the key role that the FIR plays in determining the Earth's energy budget, the information that may be contained in the spectrum, and the

lack of available measurements, Turner et al. (2015, hereafter T15) describe a methodology designed to estimate FIR radiances by exploiting correlated behaviour in the MIR. They applied this method to nadir radiance measurements from the Infrared Atmospheric Sounding Interferometer (IASI, Clerbaux et al., 2009) and evaluated their approach by comparing spectrally integrated radiances across the infrared with measurements from the Clouds and the Earth's Radiant Energy System (CERES, Wielicki et al., 1996) broadband radiometers taken during simultaneous nadir overpasses. Overall mean broadband agreement

is encouraging but the evaluation technique precludes the identification of any compensating biases within the FIR itself and is limited to polar regions.

REFIR-PAD has been measuring spectral downwelling longwave radiances at Dome-C Antarctica since 2011, providing a long-term database covering the spectral range from 100 to 1400 $cm^{-1}$ (Palchetti et al., 2015). In this study we exploit the availability of these observations to test whether a similar methodology to that described by T15 can be developed and applied

to the REFIR-PAD measurements. Here we focus on clear-sky conditions, essentially providing a test of the unique information contained in the FIR relating to water vapour spectroscopy and concentration. Because spectrally resolved observations covering much of the infrared are available, inferred FIR radiances can be compared to the real observations, providing a thorough evaluation of the success of the technique. Radiative transfer simulations utilising radiosonde measurements of the atmospheric state can also be used to assess the impact of instrumental and sampling noise on the robustness of the relationships seen. In this way we are able to assess to what level it is possible to use information in the MIR (within the constraint of the REFIR-

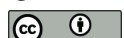


PAD wavenumber range) to infer FIR spectral behaviour using actual observations. This endeavour is particularly timely given the recent selection of the Far infrared Outgoing Radiation Understanding and Monitoring (FORUM) concept (Palchetti et al., 2016) as an ESA Earth Explorer 9 candidate mission.

In section 2, the instrumental data are described along with the radiative transfer model used to produce simulated spectra for comparisons. We also describe the distinct steps of the spectral extension method. Section 3 displays the results, with comparison between instrumental and theoretical extensions, which are discussed in section 4. We also investigate the impact of spectral averaging, consistent with the type of resolution currently employed in global climate models as a key potential use of such data are for model evaluation. Finally we draw conclusions in section 5.

## 2 Data and methodolgy

### 2.1 REFIR-PAD

The REFIR-PAD instrument is currently located at the Italian-French Concordia research station in Antarctica (75°06'S, 123°23'E) at 3,230 m above sea level. It was installed in the Physics Shelter, south of the main station buildings for the PRANA project (Proprietà Radiative dell'Atmosfera e delle Nubi in Antartide), financed by the Italian PNRA (Progetto Nazionale per la Ricerca in Antartide). The project aims to supply the first complete dataset of the spectral downwelling longwave radiances over a polar region, and has been recording data autonomously since 2011 (Palchetti et al., 2015) and within later project CoMPASS (COncordia Multi-Process Atmospheric StudieS), the currently active DoCTOR (DOme C Tropospheric ObserveR) and FIR-CLOUDS (Far Infrared Radiative Closure Experiment For Antarctic Clouds). A protective chimney separates the instrument from the outside temperature and the ingress of wind and snow is prevented by a barrier on the rooftop.

The instrument, fully described in Bianchini et al. (2006), is composed of a Fourier transform spectroradiometer (Mach-Zehnder type) with an operating spectral bandwidth of 100 - 1400 $cm^{-1}$ (100 - 7.1 $\mu$m) at a resolution of 0.4 $cm^{-1}$ and with an acquisition time of 80 s. One calibrated spectra is based on the average of four zenith observations for an overall measurement time of 6.5 min every 14 min. The noise equivalent spectral radiance (NESR) due to detector noise is approximately 1 mW $m^{-2}$ $sr^{-1}$ $(cm^{-1})^{-1}$ at 400 $cm^{-1}$. In addition to the radiometric NESR, the calibration error and the standard deviation of the four observations composing the calibrated spectrum are calculated. The standard deviation is a posteriori estimation that includes the NESR and possible scene variations (Palchetti et al., 2015).

In the study, only clear-sky cases from 2013 are used. The selection of the clear-sky spectra uses the classification outlined in Rizzi et al. (2016) to discriminate between clear and cloudy scenes. Twenty-four spectral intervals are selected and seven tests are applied, comparing the mean radiances, the standard deviation and the brightness temperature in the specified spectral intervals. This approach yields 5126 clear sky spectra for 2013. An example of a clear-sky spectrum is displayed in figure 1 and shows unphysically high radiances and standard deviations in two bands within the atmospheric window region, from 1095 - 1140 $cm^{-1}$ and 1230 - 1285 $cm^{-1}$. These are a manifestation of absorption by the polyethylene terephthalate (Mylar) substrate which composes the wideband beam splitter and hence radiances within these bands are not used in this study. Outside these two regions and where the downwelling signal is typically high (below 400 $cm^{-1}$ and between 600 and 800 $cm^{-1}$), the standard



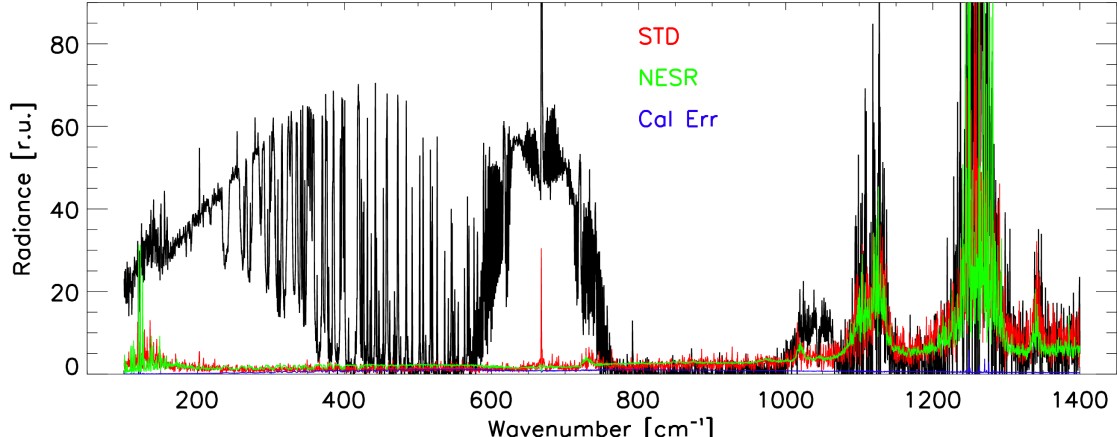

**Figure 1.** Example of a clear-sky spectrum as seen from REFIR-PAD in black, and its associated standard deviation (in red), the noise equivalent spectral radiance (in green) and the calibration error (in blue).

deviations are relatively small. However, in the most transparent regions, where the radiance is low (micro-windows between 400 - 600 $cm^{-1}$ and in the atmospheric window from 800 - 1000 $cm^{-1}$ for example), the standard deviations can exceed the measured radiances with values around 2 r.u. (radiance unit), where 1 r.u. is equivalent to 1 mW $m^{-2}$ $sr^{-1}$ $(cm^{-1})^{-1}$.

## 2.2 Radiosonde profiles

Since 2005, the radiosonde system routinely operative at Dome C has provided atmospheric pressure, temperature and humidity profiles at 12 UTC. From 2009 onwards these observations have been made using the Vaisala RS-92SPGW. The daily profiles are available at www.climantartide.it.

Data are recorded every 2 seconds, corresponding to around 800 measurements in the troposphere, and between 900 to 1900 measurements in the stratosphere, reaching up to 26-30 km (Tomasi et al., 2011). However, the relative humidity is only measured up to 15 km. Due to the balloon ascent rate (5-6 m $s^{-1}$) and the recording rate, the vertical resolution is about 10-12 m. Raw water vapour profiles are provided in relative humidity and the conversion to mixing ratio assumes saturation over water as advised by the World Meteorological Organisation (WMO) guide to meteorological instruments and methods of observation (https://www.wmo.int/pages/prog/www/IMOP/CIMO-Guide.html).

## 2.3 LBLRTM

We use the Line-By-Line Radiative Transfer Model (LBLRTM) developed by Clough et al. (2005) to simulate the downwelling radiance. The version used in this study is LBLRTM v12.7, with an updated line parameter database AER version 3.5 (following HITRAN 2012, Rothman et al. (2013)) and a continuum code MT_CKD_3.0 which includes modifications to the $H2O$ foreign





continuum in the FIR band from 0-600 $cm^{-1}$. The radiosonde profiles described in section 2.2 provide the temperature and water vapour inputs for the radiative transfer simulations. The radiosonde profiles are interpolated onto 100 levels, with the highest vertical resolution being 26 m near the surface. Additional levels extending up to 50 km in altitude are included using temperature and humidity data from the closest ERA-Interim (Dee et al., 2011) profiles in space and time, scaled to the highest

altitude where reliable temperature and water vapour values were recorded by the given radiosonde. Ozone concentrations are extracted from the same ERA-Interim profile. Minor species are taken from the AFGL sub-Arctic winter and summer profiles (Anderson, 1986) and $CO_2$ has been scaled to 2013 values as reported by NOAA's Global Monitoring Division, Earth System Research Laboratory (https://www.esrl.noaa.gov/gmd/ccgg/trends/). To achieve consistency with the REFIR-PAD instrumental characteristics, each simulated spectrum is Fourier transformed and a maximum optical path difference of 1.25 cm is applied

in the interferogram domain. The truncated interferogram is then re-transformed and the resulting spectrum is sampled at the REFIR-PAD sampling frequency.

## 2.4   Extension methodology

Based on the methodology developed by T15, FIR wavenumbers between 100 and 667 $cm^{-1}$ are correlated with (predictor) wavenumbers from 667 to 1400 $cm^{-1}$. The estimated radiance in the FIR $I_{\nu,FIR}$ can be written as a function of the predictor

radiance $I_{\nu,predictor}$, and two regression coefficients, $a_0$ and $a_1$, using:

$$\ln\left(I_{\nu,FIR}\right) = a_0 + a_1 \ln\left(I_{\nu,predictor}\right) \tag{1}$$

given the assumption of a logarithmic relationship between the predictor and estimated radiances and

$$I_{\nu,FIR} = a_0 + a_1 I_{\nu,predictor} \tag{2}$$

for a linear assumption.

20       We start by selecting the REFIR-PAD spectra that will be used to calculate the regression coefficients. All clear-sky spectra that are closest in time to the daily radiosonde measurement at 12 UTC are selected. If the closest spectrum on a given day is measured more than two hours before or after 12 UTC, the spectrum is discarded. 125 days during 2013 are retained using this criterion. These spectra are randomly divided into two sets. The first set is used as a creation set, from which the regression coefficients are derived and the second is used as a test set, on which the regression coefficients derived from the creation set

are tested.

To choose the predictor wavenumbers, we select a FIR wavenumber and create a vector composed of all radiances in the creation set at this wavenumber. We compute the correlation of this vector with a similar vector at a MIR wavenumber. We repeat this analysis for all MIR wavenumbers and select the MIR wavenumber that shows the highest correlation as the predictor for the given FIR wavenumber. Finally, the linear (or logarithmic) regression coefficients are calculated. The whole process is repeated for each FIR wavenumber. We emphasize that the methodology described here is only based on analytical considerations with the computation of the correlation. No spectral assumptions are made and as a consequence the MIR

5    predictor wavenumbers can be associated either with, for example, a $CO_2$ line, a $H_2O$ line or a combination of both.





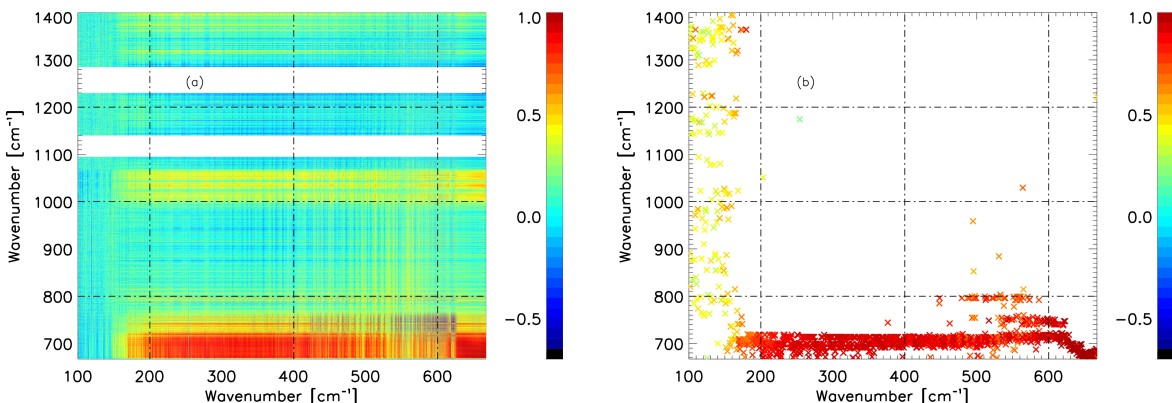

**Figure 2.** (a) Correlation map using REFIR-PAD clear-sky spectral radiances with MIR wavenumbers on the y axis and FIR wavenumbers on the x axis. (b) MIR wavenumbers that maximise the correlation for each FIR wavenumber. Colour scales indicate the correlation.

## 3 Results

### 3.1 Application to observed spectra

Figure 2(a) displays the correlation between FIR and MIR radiances using the REFIR-PAD creation set, displayed at the nominal instrument spectral resolution of 0.4 $cm^{-1}$. As noted previously, the bands corresponding to regions of high noise due to the absorption by the beam-splitter (1095 - 1140 $cm^{-1}$ and 1230 - 1285 $cm^{-1}$) have been removed from the analysis.

We observe specific spectral regions that maximise the correlation. A large portion of the spectral region between 150 and 500 $cm^{-1}$ is highly correlated with wavenumbers between 667 and 720 $cm^{-1}$. Figure 2(b) indicates the MIR predictor wavenumber selected for each FIR wavenumber using the approach described in section 2.4. Below 150 $cm^{-1}$, the predictor wavenumbers are scattered between 700 and 1400 $cm^{-1}$ with low correlation values, between 0.2 and 0.5. This can be explained by a high NESR at the edge of the REFIR-PAD detector as seen in figure 1 in green. Between 200 and 470 $cm^{-1}$, the predictor wavenumbers are clustered around 700 $cm^{-1}$, within the wing of the 667 $cm^{-1}$ $CO_2$ band, with correlations of between 0.84 and 0.94. Figure 2(a) shows lower, more varied correlations in the 470 and 570 $cm^{-1}$ region: here the MIR predictor wavenumber also shows more variability taking values varying between 696 and 799 $cm^{-1}$. The correlation values in this region lie between 0.71 and 0.92. Moving towards the centre of the 667 $cm^{-1}$ $CO_2$ band, MIR predictors are typically clustered at 682 $cm^{-1}$ and show a narrower range of higher correlations of between 0.96 and 0.98. We note that the predictor wavenumbers are mainly localised in a spectral area dominated by the $CO_2$ band coexisting with typically weaker vapour lines.

As noted in section 2.4, regression coefficients $a_0$ and $a_1$ in equations 1 and 2 are computed between each FIR wavenumber and the corresponding predictor MIR wavenumber. Figure 3 shows an example of the relationship between a predictor wavenumber at 698.4 $cm^{-1}$ and its corresponding predictand wavenumber at 301.6 $cm^{-1}$ across all creation set spectra. Logarithmic (Eqn 1) and linear (Eqn 2) fits between the radiances are also displayed.



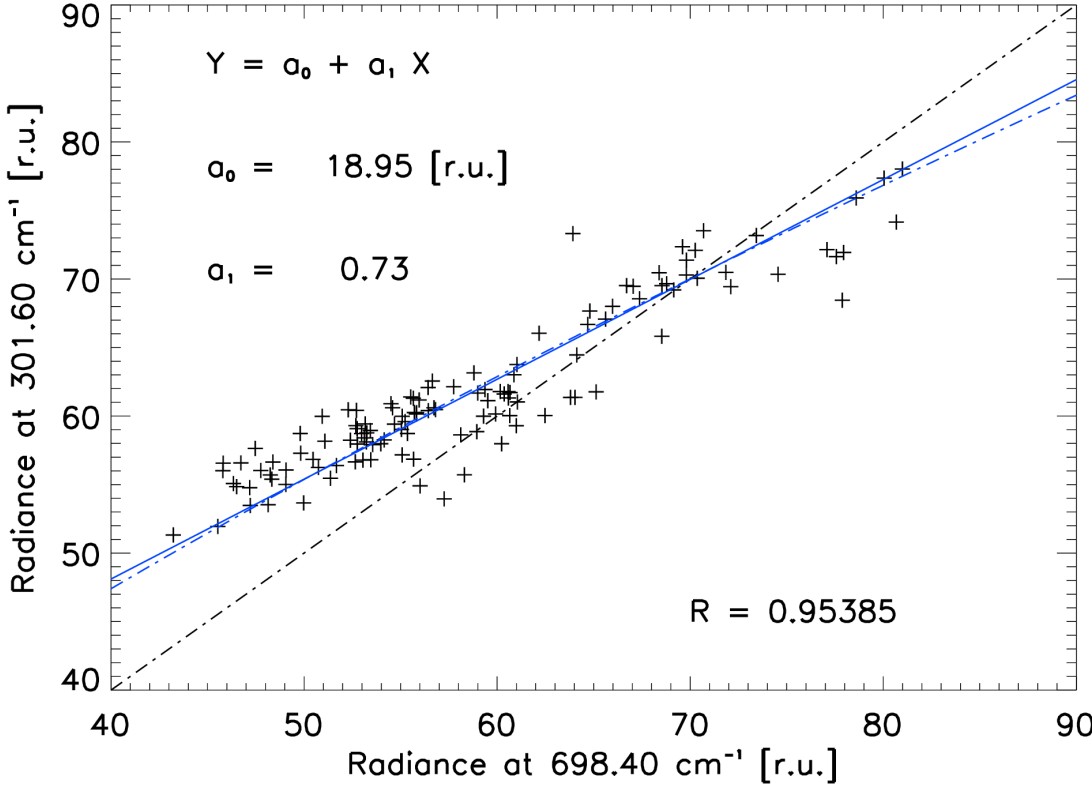

**Figure 3.** FIR radiances at 301.6 $cm^{-1}$ against predictor MIR radiance at 698.4 $cm^{-1}$ for all creation set spectra. The solid blue line is the linear fit of the points using equation 2, with regression coefficients $a_0$ and $a_1$. The linear correlation value R is indicated. For completeness, the logarithmic fit is also shown by the dashed blue line.

Using the test set of spectra we examined the robustness of the extension method. An example of a single REFIR-PAD observation (in black) and its extension (in blue) is displayed in figure 4(a) with the radiance and relative differences in figure 4(b) at 10 $cm^{-1}$ resolution. In this case, linear regressions have been used to perform all the extensions. Displaying the results at 10 $cm^{-1}$ allows a clearer picture to emerge in terms of the performance of the extension.

At 10 $cm^{-1}$ resolution the mean absolute standard error across the FIR over the entire test set is relatively small at less than 0.6 r.u. It is worth nothing that below 370 $cm^{-1}$ the mean error fluctuates around zero but at higher wavenumbers (370 $cm^{-1} < \nu$ < 600 $cm^{-1}$) there does appear to be a small positive bias of $\sim$ 0.5 r.u. Under clear-sky conditions this region is generally more transmissive than the lower wavenumber regime - as evidenced by the comparatively lower radiances in figure 4(a). These lower radiances, particularly between 400 and 570 $cm^{-1}$, contribute to slightly higher relative differences and variability across this range in figure 4(b). Above 570 $cm^{-1}$, both absolute and relative differences diminish as the radiance increases.



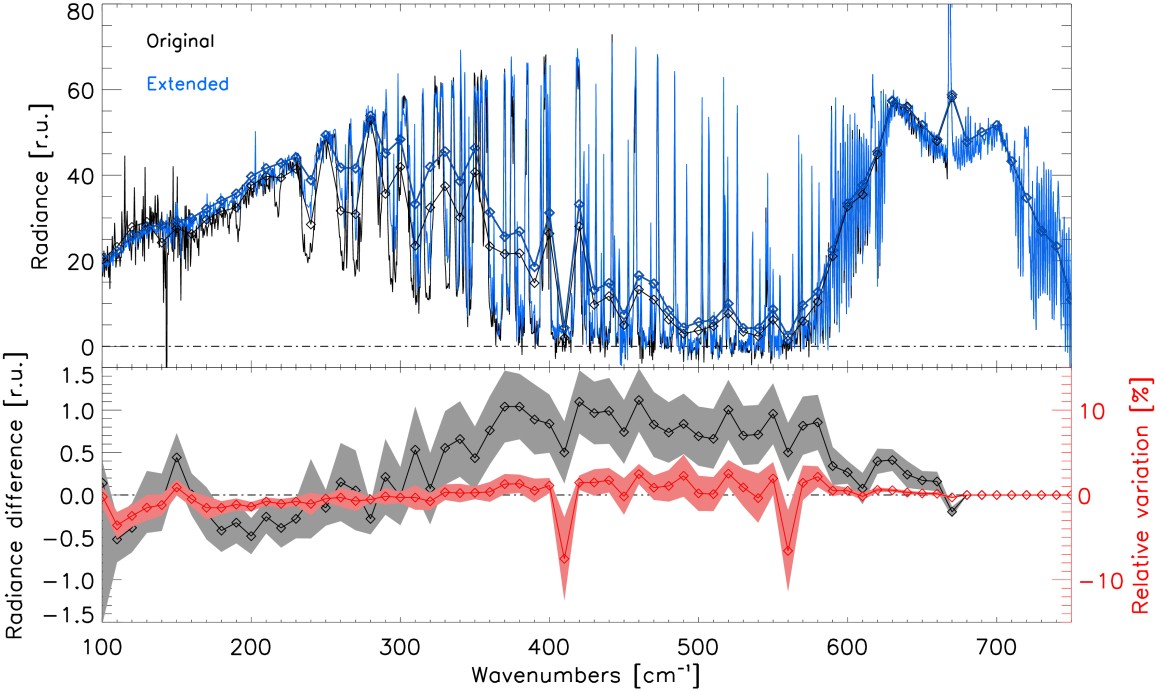

**Figure 4.** Far-infrared extension based on REFIR-PAD data. (a) Example of a spectrum (black) and its extension (blue) below 667 cm$^{-1}$. The same spectra integrated over 10 cm$^{-1}$ bands are also shown by the diamond lines. (b) Mean difference (black) and relative variation (red) between the original and the extended spectra at 10 cm$^{-1}$ resolution calculated over the entire test set. Shaded areas are the associated standard errors.

## 3.2 Application to simulated spectra

The previous section suggests that a reasonable reconstruction of observed clear-sky downwelling FIR surface spectral radiances at a moderate (10 cm$^{-1}$) resolution can be obtained using simultaneous observations of MIR radiances. In this section we explore whether similar results are obtained using simulations.

10     Therefore, we apply the same process of extension using simulated LBLRTM spectra. For each clear-sky case used to build the creation and test sets for REFIR-PAD data, the corresponding radiosonde profile is selected and used as input for LBLRTM as described in section 2.2. The output spectra are used to generate the equivalent simulated creation and test sets.

    We consider two cases. The first uses the LBLRTM spectra as directly simulated, while the second adds noise in order to be more representative of the REFIR-PAD observations. Noise is introduced using the following equation:

$I'_{\nu, LBLRTM} = I_{\nu, LBLRTM} + r * \sigma_{\nu, REFIR-PAD}$                                                                 (3)





where $I'_{\nu,LBLRTM}$ is the spectral radiance from LBLRTM with noise, $I_{\nu,LBLRTM}$ is the 'noise-free' spectral radiance directly simulated by LBLRTM, $r$ is a normally distributed random number between -1 and 1 and $\sigma_{\nu,REFIR-PAD}$ is the standard deviation from the corresponding REFIR-PAD spectrum.

The correlation maps of LBLRTM with and without noise are displayed in figures 5(a) and 5(b) respectively. Taking the no-noise case first, most wavenumbers show a strong correlation with all others, with values typically above 0.5. The FIR band sees an enhanced correlation with the MIR between 667-950 cm⁻¹ and wavenumbers between 1300 and 1400 cm⁻¹. When noise is added, the correlations reduce and show a much greater spectral variation which is more consistent with the observational case. The same bands seen in figure 2(a) which maximise the correlation appear.

The predictor wavenumbers are displayed in figure 5 without noise (c) and with noise (d). In the case of a perfect simulation, the number of predictor wavenumbers is relatively small, indicating a high degree of correlation in the spectra. Below 600 cm⁻¹, most of the predictor wavenumbers are located between 1340 and 1400 cm⁻¹. Below 300 cm⁻¹, a second band is visible around 700 cm⁻¹. Between 600 and 667 cm⁻¹, the predictor wavenumbers are spread over a range of discrete values close to 700 cm⁻¹. When the LBLRTM simulations are perturbed with noise, consistent with the change in the correlation map, the selected predictor channels show similar behaviour to REFIR-PAD (figure 2(b)). Below 160 cm⁻¹, the predictor wavenumbers are located in a wide band between 667 and 1400 cm⁻¹, but with a correlation of about 0.5. Between 200 and 400 cm⁻¹, the predictor wavenumbers are distributed in a band centred at 700 cm⁻¹. Above 400 cm⁻¹, predictor wavenumbers up to 800 cm⁻¹ also begin to appear while between 500 and 600 cm⁻¹ the spread in predictors again extends across the whole 667 - 1400 cm⁻¹ range with typically lower correlations.

At the time of writing there is only very limited spectrally resolved data in the FIR. One goal of this research is thus to see whether the LBLRTM simulations are able to provide coefficients to transfer observed MIR data into the FIR. So we now test the accuracy of going from MIR to the FIR using different approaches. All predictions are then compared against the REFIR-PAD FIR observations. The 3 different sets of regression coefficients we use are:

– LBLRTM simulations (LBL),

– LBLRTM simulations + realistic noise (LBN),

– Coupled LBLRTM (LBC) where predictor wavenumbers are generated from LBLRTM + realistic noise but regression coefficients are generated from LBLRTM without noise.

By coupling the predictor wavenumbers from LBLRTM + noise and the regression coefficients from LBLRTM without noise in the last approach, we obtain the best estimate of regression coefficients at the wavenumbers where the expected relationship is strongest.

In all cases shown a linear regression is used although the findings are essentially unchanged if a logarithmic fit is employed (see table 1). Figure 6 displays the mean differences (a) and mean relative variations (b) between the 'true' and extended FIR radiances for all cases, with their associated standard errors. For ease of comparison, the extension of REFIR-PAD based on REFIR-PAD derived regression coefficients (previously shown in figure 4(b)) is also included. The extension to the FIR





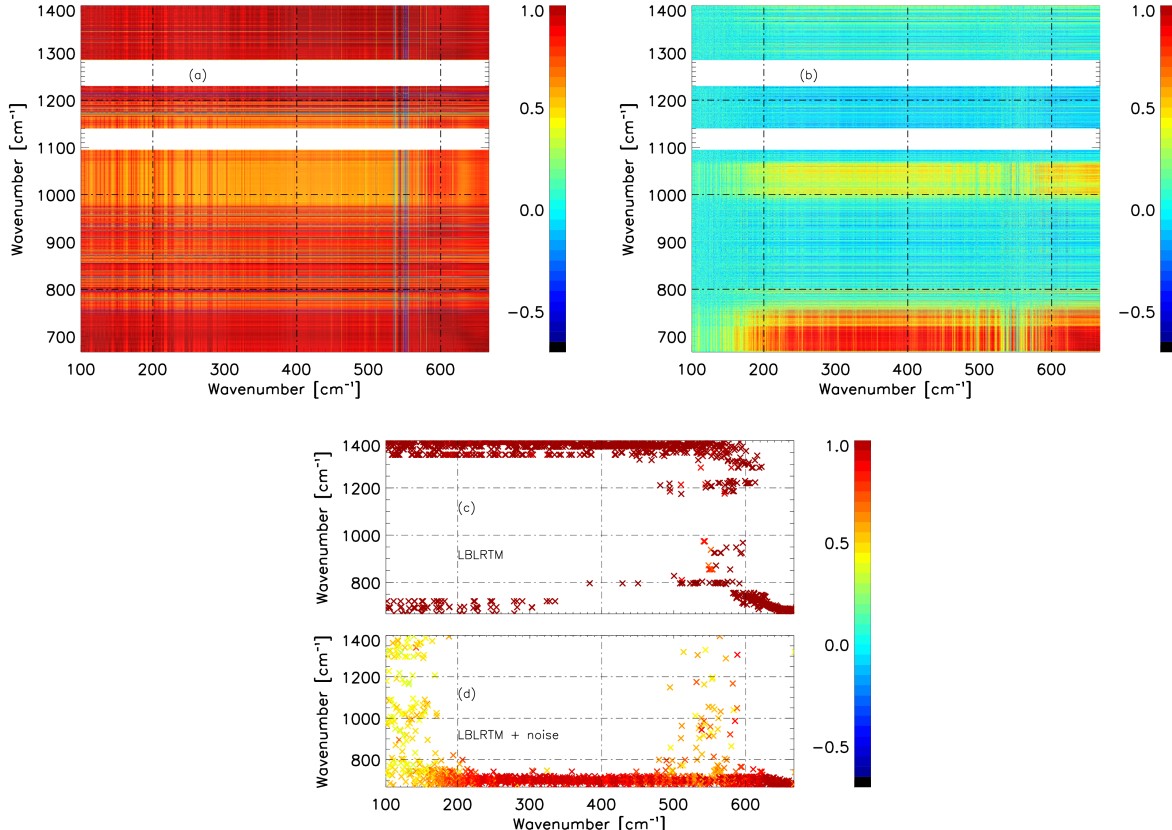

**Figure 5.** Correlation map using noiseless LBLRTM spectra (a) and LBLRTM spectra with realistic noise added (b). (c) and (d) As figure 2(b) showing predictor wavenumbers for LBLRTM without noise and with noise respectively.

using regression coefficients based on noise-free simulations (LBL) fails to capture the observed FIR behaviour. A strong bias is visible with a mean difference of -45 %. In this case, the selected predictor wavenumbers are close to 1400 cm$^{-1}$ (figure 5(b)), however, at these wavenumbers, the observed correlation for REFIR-PAD is very low (figure 2(a)), due to increased noise (figure 1), leading to large differences between the extended spectra and observations. If the predictor wavenumbers are selected from the noise adjusted simulations (LBN and LBC), the mean differences and standard errors show a marked decrease, reducing the mean difference to 1.1 % and -0.4 % for LBN and LBC respectively.

## 4 Discussion

Noise-free simulations of downwelling spectrally resolved clear-sky radiances over Antarctica imply a high level of correlation between the MIR and FIR. However, the prediction model based on these simulations fails to adequately capture observed behaviour under clear-skies as exemplified by REFIR-PAD. Instrumental noise characteristics strongly effect the choice of







**Figure 6.** As figure 4(b) for all cases of extension, with (a) the radiance difference and (b) the relative variation, using ◇ for REFIR-PAD (RFP), + for LBLRTM (LBL), $x$ for LBLRTM with noise (LBN), △ for coupled LBLRTM (LBC).





predictor wavenumbers. Including the effects of this noise in the simulations markedly improves the prediction model, which is capable of capturing the observed mean radiance in the FIR to within 2 %, except in selected bands where the downwelling radiance is low (for example 410 cm$^{-1}$, 490 cm$^{-1}$, with a peak at 540 cm$^{-1}$, see figures 6(a) and (b)).

More specific to this study, it is worth noting that the temperature and water vapour profiles very close to the ground (within 2 m) may also be affected by the presence of the chimney connecting the physics shelter to the outside environment. Palchetti et al. (2015) perform a least square minimisation of the radiance differences between the observation and the simulation, with the addition of a first level inside the chimney into the fitted profiles. Rizzi et al. (2016) include a first level inside the chimney based on the average between the internal PAD temperature and the shelter temperature. We find that the vertical resolution and assumptions made in our modelling approach are sufficient to reduce radiance biases to within 2 r.u., consistent with Rizzi et al. (2018).

In this study, the extension of REFIR-PAD has been performed on its native grid ($\Delta\nu$ = 0.4 cm$^{-1}$) and the results have been predominantly presented over averaged bands of $\Delta\nu$ = 10 cm$^{-1}$. At present, climate and Earth-system models do not operate at such a high spectral resolution. It is thus of interest to investigate how the differences presented in figures 4 and 6 are affected by integration over the wider spectral bands more typical of these general circulation models. As an exemplar, we consider the Met Office Unified Model (UM). In the UM, there are three bands with FIR contributions, from 1 - 400, 400 - 550 and 550 - 800 cm$^{-1}$.

The extensions of REFIR-PAD using the various prediction models described in section 3.2 were integrated over these bands and the corresponding results are shown in table 1. For each band and each case, the median value of the variations is provided along with the one sigma standard deviation of the relative variation across spectra in the test set. Because of the boundaries of the extension, the bands from 1 - 400 cm$^{-1}$ and 550 - 800 cm$^{-1}$ are reduced to 100.4 - 400 cm$^{-1}$ and 550 - 667 cm$^{-1}$ respectively.

Using the REFIR-PAD prediction model, integrating over wide spectral bands results in relatively small differences between the observed and extended spectra, below 3 %. However, as described earlier, the extension using simulated noise-free regression coefficients leads to strong biases, with maximum percentage differences (up to -119 %) seen in the 400-550 cm$^{-1}$ region, the most transparent of the three bands and hence the most susceptible to noise due to the low radiance level. When looking at LBN and LBC cases, the extension shows median biases which are only marginally larger than those seen using the observations themselves. In addition, the difference between using a linear or logarithmic extension is small.

## 5 Conclusions

In this study we have used REFIR-PAD downwelling observations for clear-sky cases from 2013 over Dome C in Antarctica to assess whether it is possible to build a model capable of using MIR radiances to predict values in the FIR. We have described a correlation and regression based methodology based on Turner et al. (2015) which we have used to search for predictor wavenumbers and to extract regression coefficients at these specified wavenumbers. In addition to the observations, radiosonde soundings are used to create a corresponding simulated spectral database with the radiative transfer model LBLRTM.



**Table 1.** Distribution of the differences and relative variations between the extension and the original spectra within the three bands (100.4 - 400, 400 - 550 and 550 - 667 $cm^{-1}$) for REFIR-PAD extension itself, REFIR-PAD extension based on LBLRTM (LBL), based on LBLRTM + noise (LBN) and based on coupled LBLRTM (LBC). The first and second lines correspond to a linear and a logarithmic extension respectively. The values correspond to the median value $\pm 1\,\sigma$.

| | [100.4;400] | | [400;550] | | [550;667] | |
|---|---|---|---|---|---|---|
| | [%] | [r.u.] | [%] | [r.u.] | [%] | [r.u.] |
| REFIR-PAD | $0.2 \pm 3.9$ | $0.1 \pm 1.8$ | $0.0 \pm 13.0$ | $0.0 \pm 2.0$ | $-2.2 \pm 3.4$ | $-0.8 \pm 1.3$ |
| | $-1.2 \pm 5.9$ | $-0.4 \pm 2.3$ | $3.0 \pm 9.2$ | $0.6 \pm 1.8$ | $0.2 \pm 2.1$ | $0.1 \pm 0.9$ |
| LBL | $-.9 \pm 16.0$ | $-4.1 \pm 7.2$ | $-119.5 \pm 146.4$ | $-19.4 \pm 20.2$ | $-30.3 \pm 32.0$ | $-11.2 \pm 10.0$ |
| | $-10.0 \pm 17.1$ | $-4.1 \pm 6.1$ | $-100.5 \pm 130.7$ | $-17.5 \pm 18.3$ | $-30.9 \pm 40.7$ | $-12.6 \pm 15.7$ |
| LBN | $4.6 \pm 4.0$ | $2.0 \pm 1.7$ | $-0.3 \pm 14.8$ | $-0.1 \pm 2.2$ | $-0.6 \pm 3.7$ | $-0.2 \pm 1.5$ |
| | $3.2 \pm 4.6$ | $1.2 \pm 1.8$ | $2.9 \pm 13.9$ | $0.5 \pm 2.3$ | $-2.1 \pm 3.9$ | $-0.8 \pm 1.5$ |
| LBC | $2.3 \pm 4.9$ | $0.9 \pm 1.9$ | $-6.9 \pm 16.3$ | $-1.3 \pm 2.2$ | $-4.0 \pm 3.4$ | $-1.5 \pm 1.2$ |
| | $1.5 \pm 5.4$ | $0.7 \pm 2.1$ | $-0.4 \pm 14.2$ | $-0.1 \pm 2.3$ | $-5.1 \pm 24.1$ | $-2.0 \pm 9.9$ |

Correlation maps between the observed FIR and MIR radiances show peak correlations at wavenumbers around 700 $cm^{-1}$. Noise-free simulated spectra also show strong correlations at wavenumbers between 1340 and 1400 $cm^{-1}$. With the addition of realistic noise to the simulations the pattern of the correlation map alters and looks more similar to the one created using the REFIR-PAD observations, with reduced correlations at wavenumbers < 180 $cm^{-1}$ and between 470-570 $cm^{-1}$. This indicates that the selected wavenumbers and the associated MIR to FIR correlation are both highly dependent on instrumental noise.

Using a prediction model based solely on REFIR-PAD observations, the extension from the MIR to the FIR works satisfactorily, with mean relative variations below 5 % over most of the spectral range. Between 400 and 570 $cm^{-1}$, where the atmosphere is highly transparent and the downwelling radiances are very low, the relative variation can reach up to 10 % but the absolute variation is of the same order to the rest of the spectrum (0.5 r.u.). Using a prediction model based on noise-free simulations, the extension to the FIR shows markedly poorer fidelity with the observed behaviour. However, when we add realistic instrument noise to the simulations the prediction model is able to satisfactorily estimate the REFIR-PAD FIR measurements. Where the radiance is low, higher relative differences can arise. Notable differences are also seen at wavenumbers below 150 $cm^{-1}$ which can be explained by the enhanced instrument noise close to the edge of the REFIR-PAD detector.

Our results, and those of Rizzi et al. (2018), imply that our current knowledge of clear-sky FIR spectroscopy is sufficient to match REFIR-PAD Antarctic clear-sky observations within their uncertainties given an unbiased characterisation of the atmospheric state. They also show that while it is feasible to use the type of approach we have outlined here to extend mid infrared spectral measurements to the far infrared, the quality of the extension is strongly dependent on the noise characteristics of the observations. This in turn implies that if a similar approach is developed to extend existing mid infrared ground or satellite based observations, the instrument noise must be explicitly accounted for in building the model due to its potential





role in altering the choice of predictor wavenumbers from the noise-free case. In addition, the quality of any extension using this type of method will also be critically dependent on whether the creation set of atmospheric profiles correctly represents the conditions which are actually sampled by the MIR instruments.

An obvious next step for this work would be to include cloudy conditions in the approach. However, this is challenging,
as, given the results here, one would anticipate that a good knowledge of cloud microphysics, optical properties as well as vertical location, including any impact on the associated temperature and water vapour profiles, would be required to perform the forward modelling with the requisite accuracy. The frequency of radiosonde ascents at Concordia preclude knowledge of the last effect. Cloud microphysics are not measured directly, cirrus bulk optical properties are poorly constrained in the FIR (Baran et al., 2014) and previous campaigns highlight the difficulty in matching radiance measurements across the infrared in
the presence of cirrus cloud (Cox et al., 2010). However, retrievals directly from the REFIR-PAD measurements themselves may provide a means to circumvent some of these issues in future (Di Natale et al., 2017; Maestri et al., 2018).

More generally, one would want to test this synthetic approach more comprehensively. If selected, the candidate ESA Earth Explorer 9 mission, FORUM could provide the extensive, simultaneous FIR and MIR observational database needed to validate such a prediction model. With a correct appreciation of the role of instrument noise, such a model could then be applied
retrospectively to existing MIR hyperspectral measurements to derive a long-term record of spectrally resolved radiances suitable for climate model evaluation.

*Competing interests.* The authors declare that they have no conflict of interest.

*Acknowledgements.* The analysis was supported by the NERC-funded International Consortium for the Exploitation of Infrared Measurements of PolAr ClimaTe (ICE-IMPACT) project (grant NE/N01376X/1) and by the National Centre for Earth Observation, UK. The deployment of REFIR-PAD in Antarctica was supported by the Italian National Program for Research in Antarctica PNRA (Programma Nazionale di Ricerche in Antartide) under the following projects: 2009/A04.03, 2013/AC3.01 and 2013/AC3.06. The REFIR-PAD data are available from http://refir.fi.ino.it/refir-pad-domeC. Radiosounding measurements are part of the IPEV/PNRA Project "Routine Meteorological Ob-
servation" at Station Concordia-www.climantartide.it/index.php?lang=enea. ECMWF data were acquired from http://www.ecmwf.int/en/research/climate-reanalysis/erainterim.



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
