# Peer review of "Can downwelling far-infrared radiances over Antarctica be estimated from mid-infrared information?"

_Atmospheric Chemistry and Physics, 2018_

## Referee Comment (RC1) · Anonymous Referee #1 · 14 Nov 2018

This paper is clear and well presented with a valid methodology to simulate FIR radiances from the Mid-IR. However it is not clear what is the need for this as radiative transfer models can make the same simulation with probably better accuracy.

The other major limitation is the results are only valid for radiances over Antarctica which limits the interest as to be of real value to climate modellers a more global application of this technique is required.

Hence before accepting publication of this paper I would like to see a much clearer explanation for why this technique would be useful given its limitations outlined in the paper. Also a demonstration of the technique over a wider range of atmospheric con-

ditions is needed.

---

## Referee Comment (RC2) · Anonymous Referee #2 · 20 Nov 2018

Overall impression

The authors note, correctly, the sparsity of far-infrared hyperspectral radiance measurements compared to the mid-infrared. Given the important role of the far-infrared in climate and energy budget studies it is a worthwhile endeavour to seek to improve our knowledge of the full infrared spectrum. The methodology of Turner et al. (2015), which entails using mid-infrared predictor channels to infer far-infrared spectral radiances, is applied to a dataset of Antarctic downwelling longwave radiances (REFIR-PAD). The paper tests whether the methodology can successfully extend/extrapolate mid-infrared spectra into the far-infrared.

[Figure]

Within certain conditions the authors show reasonable skill in predicting the FIR spectrum from MIR data. Notwithstanding some points about how this works in practice (see below) I think the general idea has applicability in climate science where FIR data are lacking. The authors note that there is renewed interest in the remote sensing community in spectrally resolved FIR measurements (specifically the candidate ESA mission FORUM). I hope the authors can address the more detailed points raised below.

Substantive points

1. Page 3, line 27. "In the study, only clear-sky cases from 2013 are used." There may be good reasons for this choice, but these are not clear. Please explain why this data selection was made.

2. Methodology, page 5. There are various differences between the published method (T15) and the way it is applied here. In T15 IASI data (upwelling radiances) are used cf. downwelling REFIR-PAD data; T15 data are restricted (it seems, on quick read) to between 30S and 60N cf. cold/dry Antarctic conditions here; T15 find maximum correlations either side of peak water vapour absorption at 1600 cm-1 while REFIR-PAD is restricted to below 1400 cm-1. These are not commented upon. In particular I would like the authors to comment on the general applicability of their scheme for global climate studies when this paper deals exclusively with polar conditions.

3. The method relies on finding a single predictor MIR wavenumber for each FIR wavenumber. This is straightforward and as published in T15. However, there may be potential downsides to using single frequencies, particularly when the same predictor is selected multiple times and in cases of instrument noise. Have the authors considered whether a weighted average or linear combination of the most highly correlated channels might work better as a predictor?

4. Some plots (Figs 4, 6) show standard errors to represent the uncertainty. I am uncomfortable with this, as it implies the expected error in the method can be reduced to the mean error with an arbitrarily large sample. The uncertainty is better represented

as the example in Fig. 4 (a) where the difference between spectrum and extension in microwindows near 240/260/270 cm-1 is about 10 r.u. Elsewhere (Table 1) you quote one standard deviation which seems more appropriate to me.

5. Page 8, Eq. (3). You add noise according to the REFIR-PAD standard deviation, but you discount correlated errors e.g. the calibration error in Fig. 1. If you want to represent the instrument uncertainty shouldn't this be included?

6. You show (Fig. 5) very different choices of predictor channels when using noiseless and noise-added LBLRTM spectra (the latter look very like those found from REFIR-PAD in Fig. 2). You say in the discussion and conclusions that including the effects of noise is important to improve the prediction model. But I think this avoids the interesting question of why the noise-free LBL extension fails (but why LBN and LBC seem to work). Plausibly it comes down to the choice of predictor channels which are heavily skewed towards 1400 cm-1 in the LBL case. Just because channels are correlated doesn't mean they have a linear relationship. It might be interesting to see an equivalent of Fig. 3 for an LBL case.

7. If instrument (random) noise is critical in your method there may be benefit in noise filtering the measured spectra. Methods exist for doing this, e.g. via principal component analysis. I anticipate this may be beyond the scope of the present study but may be worth pursuing in future work.

Minor points

1. Page 2, line 9. Typo "vvvhave".

2. Page 3, line 22. "One calibrated spectra", strictly "spectrum".

3. Fig. 3, I assume the dot-dashed line is a 1:1 line but this is not stated.

4. Page 10, line 8, you refer to "figure 5 (b)" when it looks like (c)?

---

## Referee Comment (RC3) · Anonymous Referee #3 · 26 Nov 2018

Overall comments

The paper "Can downwelling far-infrared radiances over Antarctica be estimated from mid-infrared information?" demonstrates a method for estimating downwelling far-IR spectral radiances at the surface in Antarctica from mid-IR radiances (both measured and calculated) at that location. Although the main result in the paper is adequately presented, overall the paper suffers from significant motivational and methodological issues.

This study is presented as similar to the TOA-based study of Turner et al. (2015). The need for the results presented in the earlier study can be straightforwardly seen. There

exist many millions are mid-IR spectral radiance measurements from satellites (IASI, CrIS, AIRS, TES) without far-IR counterparts, and the construction of far-IR simulated radiances consistent with the mid-IR observed radiances could be very useful. That these observations are global and span more than a decade also attest to their utility. However, the same is not the case for surface radiances. This paper does not really attempt to make an argument that the development of this technique is likely to be relevant or the lessons learned from this study will be able to be applied to something relevant. There is no global set of mid-IR ground-based spectral radiometers for which this technique would be useful. The closest is the many AERIs that have been deployed, but the vast majority are located in regions with high enough water vapor so the far-IR is opaque and a simple surface air temperature measurement will suffice to predict the far-IR radiances. Only the few AERI datasets from high-latitude or -altitude might be spectrally extended using a variant of this technique, but even that application begs the question as to what would be the need to adequately fill in the far-IR record for these deployments. A weak motivational argument is made by alluding to the FORUM mission, but this is a satellite-based instrument. Without some adequate motivation, there is no real reason for this paper to be published.

The methodology employed is also problematic. In a dry location like Antarctica, far-IR radiances are primarily determined by the temperatures and water vapor amounts in the lowest layers of the atmospheres. Therefore, a technique to predict far-IR radiances using a single observed mid-IR radiance would ideally use a frequency that also is sensitive to these atmospheric conditions. However, the paper indicates that any such REFIR-PAD channels (1300-1400 cm-1) are too impacted by noise to be very useful for this application. Therefore the method that is arrived at predominantly uses frequencies in the CO2 band, measurements that basically are sensitive just to the temperatures at the levels very close to the instrument. (The text incorrectly suggests that water vapor is also important in this spectral region.) Therefore, this result in the paper can be boiled down to most important consideration in simulating far-IR radiances, given the noise in REFIR-PAD mid-IR water vapor channels, is getting the near-instrument temperature

correct. Understanding the result in this context should give the authors some pause in any assessing the importance of this result – if the simulated far-IR radiances are independent of the water vapor profile, how useful and information-laden can they be?

Given the ease these days of doing non-simple optimization, it's unclear why the authors did not determine an optimal linear combination of the radiances at multiple channels for each far-IR radiance. By definition, this would obtain better results than using a single channel and allow the use of spectral points sensitive to both temperature and water vapor.

The comments above mainly apply to the results in the paper about using observed mid-IR radiances to simulate far-IR radiances. The paper also has results for using calculated mid-IR radiances to obtain far-IR radiances. It's not clear if these results would be useful since if one were already doing calculations and wanted far-IR radiances, why not compute them directly?

Specific comments by page, line. More important comments denoted by *:

Section 1

2, 3 – perhaps use "as much of as N %" instead of "significant" since "significant" is pretty subjective (and the fraction is probably less than 50%)

2,9 – some extra "v" characters are present

2, 13 – It might be worth mentioning the RHUBC campaign (papers by Turner, Mlawer)

3,1-3 – A potential satellite mission is not relevant to this paper's purpose since this analysis only applies to ground-based measurements. It may be relevant to the T15 paper, but not to this paper.

Section 2

3,15 – The REFIR_PAD goes up to only 1400 cm-1, so it's hard to see how that could be a "complete longwave dataset."

4,9 – Perhaps start this sentence "Each radiosonde records data every 2 seconds...". The current wording might be taken as sondes are launched every 2 seconds.

4,19 – Perhaps get rid of "developed by" and change Clough et al. into a regular reference (i.e. in parentheses). The current wording makes it seem like the model was developed around 2006.

5,1 – In the far-IR, the linefile has substantial modifications to the HITRAN 2016 widths following Delamere et al. and Mlawer et al. The latter study also led to MT_CKD_3.0. The text says "includes modifications", but doesn't specify what the modifications were in reference to (i.e. modifications to what?). Since the RHUBC-II results are primarily based on REFIR-PAD measurements, it is probably worth mentioning when introducing the model.

5,22 – Do the conditions have to be clear for the entire period between when the sonde is launched through the REFIR-PAD measurement time?

5,25 – random subsampling?

Section 3

6,8 and Fig. 1 – Specify whether these results are for linear or log.

*9,2 – The noise added is between -1 and 1 times the standard deviation of the measured noise at each frequency. Doesn't this underestimate the actual noise? Why not use the chosen random number to sample from a normal distribution with that standard deviation? Also, does the actual noise have spectral correlation? (i.e. if a case is higher than the mean in the MIR, is it likely to be higher than the mean in the FIR?). If so, then not taking that into account in the method may lead to inferior results for the LBLRTM+noise compared to using a pure measurement approach.

9, 25-29 – It should be made more clear that all these LBLRTM-based regression approaches are being applied to the MIR observations and not the LBLRTM simulated radiances.

10,11 – Using the mean difference may allow some cancellation of errors between the spectral points that are being averaged. It might be better to use the mean absolute value of the differences.

Section 4

10,15 – "exhibited" would be better than "exemplified". Also, "affect", not "effect".

*11, Fig 6 – I think that the result that is plotted is from a single case. If so, please label it as such. However, if true, that opens up a more serious critique. Until the paper gets to Table 1, all results shown and discussed are from an example, not from the full dataset. How can the reader know whether these results are representative of the entire dataset?

*12,9-15 – This section is puzzling. It is not up to a user's discretion whether to interpret the relative humidity measured by a sonde as being with respect to liquid or ice saturation pressure. This is determined by the sonde design and processing software, and is done with respect to liquid. Interpreting it with respect to ice is not correct. In addition, it is difficult to understand the logic behind the statement in lines 13-15 "(indirectly implies . . ."). Why would changing the poor results obtained from sonde water vapor profiles obtained by the method described in this paragraph have anything to do with applying the methods in this paper to other conditions? This entire paragraph should be deleted.

12,21 – Perhaps add a few words to clarify: ". . . the vertical resolution and assumptions made in our modelling approach without adding a chimney layer are sufficient . . ."

Section 5

*14,1 – The Rizzi et al. paper certainly shows that current spectroscopy is sufficient to match observations and is an improvement over previous spectroscopy. The results in this paper show nothing of the kind. Perhaps the LBLRTM results the authors performed also indicate this. However, these results have not been presented in this

manuscript.

*14,2-3 – The paper has not shown that an unbiased atmospheric state is essential to the approach that has been demonstrated. Water vapor profiles from sondes under very dry conditions are known to be biased (e.g. Miloshevich et al. (2009)), so the profiles that are used in this paper are likely far from unbiased.

*14,13-15 – As in the comment above, this really has not been shown. At best, the analysis about the water vapor saturation pressure over ice that is alluded to (but not really shown) suggests that this might be true, but this is far from being demonstrated.

14,24-29 – It is unclear what the results from this ground-based study have to do with the possible future FORUM mission. The authors should make their argument here more clearly or abandon it.

---

## Author Comment (AC1) · 6 Feb 2019

Referee #1 - 01: This paper is clear and well presented with a valid methodology to simulate FIR radiances from the Mid-IR. However it is not clear what is the need for this as radiative transfer models can make the same simulation with probably better accuracy.

Response 01: One point that needs to be made here is that radiative transfer tools have not been extensively validated in the far infra-red. The evaluation which has been done is limited by the available measurements covering this spectral region, which are not extensive – indeed the REFIR-PAD dataset is the only such dataset covering the

full FIR spectral range over an extensive (multi-season) period of time. A related question then becomes why one would attempt to extend the MIR to cover the FIR using the approach discussed here rather than simply using the data to directly evaluate the LBLRTM code over this range. We argue that several authors have described similar techniques, then actively implemented them to extend satellite mid-infrared radiances to cover the FIR (Turner et al., 2015, Huang et al., 2013, Huang et al., 2008). While we are considering a different viewing geometry, and thus might expect a different pattern of spectral correlations or patterns to emerge, in this case we are able to actually test the level of spectral agreement obtained with real observations, something not possible in the earlier studies due to the lack of space-based FIR spectral measurements. Indeed, while both earlier studies show good agreement with observations at the broadband level, comparisons between an extended AIRS dataset and GCM simulations show that such agreement could still hide compensating effects across the spectrum.

Huang, X.; Yang, W.; Loeb, N. G. & Ramaswamy, V., Spectrally resolved fluxes derived from collocated AIRS and CERES measurements and their application in model evaluation: Clear sky over the tropical oceans, Journal of Geophysical Research: Atmospheres, 2008, 113, n/a-n/a Huang, X.; Cole, J. N. S.; He, F.; Potter, G. L.; Oreopoulos, L.; Lee, D.; Suarez, M. & Loeb, N. G., Longwave Band-By-Band Cloud Radiative Effect and Its Application in GCM Evaluation, Journal of Climate, 2013, 26, 450-467 Turner, E. C.; Lee, H.-T. & Tett, S. F. B., Using IASI to simulate the total spectrum of outgoing long-wave radiances, Atmospheric Chemistry and Physics, 2015, 15, 6561-6575

Referee #1 - 02: The other major limitation is the results are only valid for radiances over Antarctica which limits the interest as to be of real value to climate modellers a more global application of this technique is required. Hence before accepting publication of this paper I would like to see a much clearer explanation for why this technique would be useful given its limitations outlined in the paper.

Response 02: The main reason why the study has only been performed over Antarctica is that, as noted above, the only instrument which has measured the complete FIR downwelling spectral radiance over an extended period of time is the REFIR-PAD located at Antarctica. We anticipate that, given the very specific meteorological conditions seen at this location the relationships we find here may well not hold for other locations but our goal is to use the existing data as a starting point to see whether the approach is actually viable, and what the key drivers of uncertainty are. This is timely because there is at least one mission (FORUM) that could provide the global observational data needed to derive such a relationship (for top of atmosphere radiances) in future. This could then be retrospectively applied with more confidence to numerous older, spectral MIR only, satellite observations. It is also worth noting that Turner et al. (2015) use only one regression relationship regardless of atmospheric state (i.e. they assume the relationship derived from a limited number of radiative transfer simulations is robust across all conditions) while Huang et al. (2008) derive clear-sky FIR estimates from MIR observations based on principal component analysis of a more extensive data set of simulated spectra, but for relatively coarse intervals of surface temperature, precipitable water and lapse rate.

Referee #1 - 03: Also a demonstration of the technique over a wider range of atmospheric conditions is needed.

Response 03: If the reviewer means a wider range of clear-sky conditions then obviously we are limited by the typical range of conditions seen at Concordia, which have effectively been sampled by considering a whole year of data. If the comment is more towards the fact that only clear-sky conditions are considered this is mainly because, as noted above, our goal is to determine whether the technique is viable, not to develop a method that can be applied to all-conditions and locations. In principle, if we see large differences at this point, for relatively constrained conditions, it is unlikely that a single set of regression coefficients will yield acceptable results in all locations, even if derived from simulations sampling all conditions. A second factor is the difficulty in discriminating between different types of cloud/precipitation (e.g. cirrus, diamond

dust, etc.) and constraining the cloud microphysics – which we would expect to have a strong influence on the observed spectral shape across the infrared. Since the approach relies on forward modelling, without this information we would have the concern that our forward simulations were not really representative of the observed conditions and hence the statistical relationships derived between the MIR and FIR could not be expected to capture the true behaviour.
* * *
Referee #2 - 04: Overall impression The authors note, correctly, the sparsity of far-infrared hyperspectral radiance measurements compared to the mid-infrared. Given the important role of the far-infrared in climate and energy budget studies it is a worthwhile endeavour to seek to improve our knowledge of the full infrared spectrum. The methodology of Turner et al. (2015), which entails using mid-infrared predictor channels to infer far-infrared spectral radiances, is applied to a dataset of Antarctic downwelling longwave radiances (REFIR-PAD). The paper tests whether the methodology can successfully extend/extrapolate mid-infrared spectra into the far-infrared. Within certain conditions the authors show reasonable skill in predicting the FIR spectrum from MIR data. Notwithstanding some points about how this works in practice (see below) I think the general idea has applicability in climate science where FIR data are lacking. The authors note that there is renewed interest in the remote sensing community in spectrally resolved FIR measurements (specifically the candidate ESA mission FORUM). I hope the authors can address the more detailed points raised below. Substantive points 1. Page 3, line 27. "In the study, only clear-sky cases from 2013 are used." There may be good reasons for this choice, but these are not clear. Please explain why this data selection was made.

Response 04: The purpose of this study is to investigate the statistical correlations between the mid-infrared and far-infrared portion of the downwelling spectra in Antarctica. This can be done by exploiting a dataset covering the seasonal variability during a whole year. The 2013 dataset was chosen since it represents one of the years with

a very large number of spectra, with a good estimate of the spectral error.

Our reason for focusing on clear-sky conditions only is as noted in the responses to reviewer 1: our goal is to determine whether a regression type extension technique can give reasonable accuracy when compared to real observations, not to develop a method that can be applied to all-conditions and locations. In principle, if we see large differences at this point, for relatively constrained conditions, it is unlikely that a single set of regression coefficients will yield acceptable results in all locations, even if derived from simulations sampling all conditions. The main variables which contribute in both the FIR and MIR portions of the terrestrial spectrum are water vapour and clouds. For this study we want to separate these two contributions, since the cloud effect needs to be investigated separately and more accurately: in particular a lack of ancillary observations means it is difficult to discriminate between different types of cloud/precipitation (e.g. cirrus, diamond dust, etc.) and constrain the cloud microphysics – which we would expect to have a strong influence on the observed spectral shape across the infrared. Since the approach relies on forward modelling, without this information we would have the concern that our forward simulations were not really representative of the observed conditions and hence the statistical relationships derived between the MIR and FIR could not be expected to capture the true behaviour.

Referee #2 - 05: 2. Methodology, page 5. There are various differences between the published method (T15) and the way it is applied here. In T15 IASI data (upwelling radiances) are used cf. downwelling REFIR-PAD data; T15 data are restricted (it seems, on quick read) to between 30S and 60N cf. cold/dry Antarctic conditions here; T15 find maximum correlations either side of peak water vapour absorption at 1600 cm-1 while REFIRPAD is restricted to below 1400 cm-1. These are not commented upon.

Response 05: Text has been added in the introduction to highlight the differences between T15 and our study. We start with the fact that T15 uses LBLRTM simulations to create the prediction model but without the possibility to verify that the prediction model is consistent with observational spectra since there are no FIR observations

from space. Therefore, the comparison of the extended spectra is also impossible to realise in T15 and only broadband validation is performed. T15 wavenumbers of maximum correlation are also recalled in section 3.2.

Referee #2 - 06: In particular I would like the authors to comment on the general applicability of their scheme for global climate studies when this paper deals exclusively with polar conditions.

Response 06: Please see our first response above and also those to reviewer 1, which highlight how our intention was not to create a generally applicable scheme but rather to test whether such a technique could generate realistic spectral radiances when compared to real observations. As such we would not necessarily expect the spectral relationships we see to hold under conditions that are substantially different to those sampled here: for example, the additional water vapour seen at lower latitudes would be expected to have a substantially greater impact in the FIR micro-windows than the main atmospheric window. We have now made this intent clearer in the revised manuscript.

Referee #2 - 07: 3. The method relies on finding a single predictor MIR wavenumber for each FIR wavenumber. This is straightforward and as published in T15. However, there may be potential downsides to using single frequencies, particularly when the same predictor is selected multiple times and in cases of instrument noise. Have the authors considered whether a weighted average or linear combination of the most highly correlated channels might work better as a predictor?

Response 07: Indeed, updating the method using a weighted average or linear combination of the most highly correlated channels could improve the results, especially at wavenumbers lower that 170 cm-1 where the correlation decreases (figure 2(b) and 5(d)). We would not expect much changes between 200 and 450 cm-1 since the correlation is higher. However, we think our approach works well enough that adding more complexity at this stage is not necessary.

Referee #2 - 08: 4. Some plots (Figs 4, 6) show standard errors to represent the

uncertainty. I am uncomfortable with this, as it implies the expected error in the method can be reduced to the mean error with an arbitrarily large sample. The uncertainty is better represented as the example in Fig. 4 (a) where the difference between spectrum and extension in microwindows near 240/260/270 cm-1 is about 10 r.u. Elsewhere (Table 1) you quote one standard deviation which seems more appropriate to me.

Response 08: Actually, the standard error as used in figure 4 and 6 stands for the standard deviation of the extension among the set of spectra considered. Therefore, the figures are consistent with table 1. It was set as standard error in order to avoid confusion between the standard deviation of a single REFIR-PAD spectrum and the standard deviation relative to the extension of a specific set. However, it has been corrected in the updated version of the article.

Referee #2 - 09: 5. Page 8, Eq. (3). You add noise according to the REFIR-PAD standard deviation, but you discount correlated errors e.g. the calibration error in Fig. 1. If you want to represent the instrument uncertainty shouldn't this be included?

Response 09: We used the standard deviation, and this does actually include the NESR in its computation. Rizzi et al. (2016) uses the maximum between CAL + NESR and $\sigma$. Our results have been updated to be consistent with MAX(SQRT(CAL$^2$+NESR$^2$), $\sigma$), as expressed with equation 3.

Referee #2 - 10: 6. You show (Fig. 5) very different choices of predictor channels when using noiseless and noise-added LBLRTM spectra (the latter look very like those found from REFIRPAD in Fig. 2). You say in the discussion and conclusions that including the effects of noise is important to improve the prediction model. But I think this avoids the interesting question of why the noise-free LBL extension fails (but why LBN and LBC seem to work). Plausibly it comes down to the choice of predictor channels which are heavily skewed towards 1400 cm-1 in the LBL case. Just because channels are correlated doesn't mean they have a linear relationship. It might be interesting to see an equivalent of Fig. 3 for an LBL case.

[Figure]

Response 10: We believe that the fact that LBL fails to extend correctly to the FIR originates from the lack of correlation in REFIR-PAD data at the wavenumbers that maximise the correlation for LBL. When comparing the correlation see in figure 5(c) and the corresponding blue/yellow area in figure 2(a), the correlations in REFIR-PAD data do not exceed 0.5.

The equivalent of figure 3 for LBL cases is shown below (figure r1), exhibiting, as expected, a much higher correlation coefficient. The predictor wavenumber here is 1376.00 cm-1. Most of the highly correlated LBL wavenumber duos follow this shape, close to a linear relationship.

Figure r1: FIR radiances at 447.20 cm-1 against predictor MIR radiance at 1376.00 cm-1 for simulated LBLRTM cases (same as figure 3).

Referee #2 - 11: 7. If instrument (random) noise is critical in your method there may be benefit in noise filtering the measured spectra. Methods exist for doing this, e.g. via principal component analysis. I anticipate this may be beyond the scope of the present study but may be worth pursuing in future work.

Response 11: Thank you for the suggestion. Indeed, this could be used in future work along with the optimisation of the predictor wavenumbers.

Referee #2 - 12: Minor points 1. Page 2, line 9. Typo "vvvhave". 2. Page 3, line 22. "One calibrated spectra", strictly "spectrum". 3. Fig. 3, I assume the dot-dashed line is a 1:1 line but this is not stated. 4. Page 10, line 8, you refer to "figure 5 (b)" when it looks like (c)?

Response 12: These have been corrected.

———————————————————————————————————————————

Referee #3 - 13: Overall comments The paper "Can downwelling far-infrared radiances over Antarctica be estimated from mid-infrared information?" demonstrates a method for estimating downwelling far-IR spectral radiances at the surface in Antarctica from

mid-IR radiances (both measured and calculated) at that location. Although the main result in the paper is adequately presented, overall the paper suffers from significant motivational and methodological issues. This study is presented as similar to the TOA-based study of Turner et al. (2015). The need for the results presented in the earlier study can be straightforwardly seen. There exist many millions are mid-IR spectral radiance measurements from satellites (IASI, CrIS, AIRS, TES) without far-IR counterparts, and the construction of far-IR simulated radiances consistent with the mid-IR observed radiances could be very useful. That these observations are global and span more than a decade also attest to their utility. However, the same is not the case for surface radiances. This paper does not really attempt to make an argument that the development of this technique is likely to be relevant or the lessons learned from this study will be able to be applied to something relevant. There is no global set of mid-IR ground-based spectral radiometers for which this technique would be useful. The closest is the many AERIs that have been deployed, but the vast majority are located in regions with high enough water vapor so the far-IR is opaque and a simple surface air temperature measurement will suffice to predict the far-IR radiances. Only the few AERI datasets from high-latitude or –altitude might be spectrally extended using a variant of this technique, but even that application begs the question as to what would be the need to adequately fill in the far-IR record for these deployments. A weak motivational argument is made by alluding to the FORUM mission, but this is a satellite-based instrument. Without some adequate motivation, there is no real reason for this paper to be published.

Response 13: As mentioned in the introduction, T15 applied the methodology to nadir radiance measurements from the IASI instrument and evaluated their approach by comparing spectrally integrated radiances across the infrared with measurements from the CERES broadband radiometers. Overall mean broadband agreement has been shown to be encouraging but the evaluation technique precludes the identification of any compensating biases within the FIR itself. Our intention here was to test whether it is possible to use such a technique to infer spectral FIR radiances from MIR radiances,

using observational spectra to validate the results obtained. Given the comments of this, and the other reviewers, we did not make it clear enough in the original manuscript that the goal was not to develop a new technique to be applied to all ground-based observations (of which we are well aware there are few) but to test the general principle of the approach.

Referee #3 - 14: The methodology employed is also problematic. In a dry location like Antarctica, far-IR radiances are primarily determined by the temperatures and water vapor amounts in the lowest layers of the atmospheres. Therefore, a technique to predict far-IR radiances using a single observed mid-IR radiance would ideally use a frequency that also is sensitive to these atmospheric conditions. However, the paper indicates that any such REFIR-PAD channels (1300-1400 cm-1) are too impacted by noise to be very useful for this application. Therefore the method that is arrived at predominantly uses frequencies in the CO2 band, measurements that basically are sensitive just to the temperatures at the levels very close to the instrument. (The text incorrectly suggests that water vapor is also important in this spectral region.) Therefore, this result in the paper can be oiled down to most important consideration in simulating far-IR radiances, given the noise in REFIR-PAD mid-IR water vapor channels, is getting the near-instrument temperature correct.

Response 14: The reviewer is absolutely correct that in an ideal world one would want to exploit observations from 1300-1400 cm-1 (or even higher wavenumbers to properly sample the water vapour rotation-vibration band) and that we are limited by the REFIR-PAD noise here. However, while we agree that for the noisy case most of the 'skill' in the extension to wavenumbers below 450 cm-1 is mainly coming from near surface temperature (via CO2 emission at around 700-720 cm-1), between 450-600 cm-1 there is certainly information from water vapour variability and profile (see figures r2 and r3 below which shows the impact of variations in profile for a typical radiosonde profile from Antarctica across the full spectral range of REFIR-PAD). We also note that at no point in the original manuscript did we state that water vapour was important in

the CO2 band wings – in fact we explicitly called them just that – however there are small water vapour features within this range (see zoomed in plot r4 and r5) and some of the predictor wavenumbers will include their contribution.

Figure r2: Differences between downwelling radiances computed using an atmospheric profile with and without water vapour (in red), and for 2 cases of temperature perturbations (+/- 1 standard deviation, respectively in blue and green, computed over the whole set of atmospheric profiles used in this study).

Figure r3: Differences between downwelling radiances with local perturbations in the water vapour profile. In red, the VMR below 5 km has been increased of 10 %, in green (blue), the VMR between 5 and 10 km (between 10 and 20 km) has been increased of 10 %.

Figure r4: Relative difference for the same cases as figure r2, between 650 and 800 cm-1.

Figure r5: Relative difference for the same cases as figure r3, between 650 and 800 cm-1.

Referee #3 - 15: Understanding the result in this context should give the authors some pause in any assessing the importance of this result – if the simulated far-IR radiances are independent of the water vapor profile, how useful and information-laden can they be? Given the ease these days of doing non-simple optimization, it's unclear why the authors did not determine an optimal linear combination of the radiances at multiple channels for each far-IR radiance. By definition, this would obtain better results than using a single channel and allow the use of spectral points sensitive to both temperature and water vapor.

Response 15: As noted above (and shown by the figures) there is clearly sensitivity to the water vapour profile in the far-infrared and also, increasingly from wavenumbers from around 710 cm-1 and above, even for this ground-based viewing geometry.

We do appreciate that including multiple channels in the predictor approach may well have been beneficial (noise permitting) and this is now explicitly stated in the revised manuscript and would be an avenue to pursue in future when developing a more generic approach.

Referee #3 - 16: The comments above mainly apply to the results in the paper about using observed mid-IR radiances to simulate far-IR radiances. The paper also has results for using calculated mid-IR radiances to obtain far-IR radiances. It's not clear if these results would be useful since if one were already doing calculations and wanted far-IR radiances, why not compute them directly?

Response 16: Essentially this is the same point as one raised by reviewer 1. If one had complete confidence in the ability of radiative transfer codes to correctly simulate far-infrared radiances then of course one could use them across the whole infrared range. In practice we simply do not have this confidence (as evidenced by the numerous updates to water vapour spectroscopy in this region over time as new observations become available) due to the very limited number of measurements. Moreover, even if we had complete confidence, as the reviewer themselves state 'There exist many millions are mid-IR spectral radiance measurements from satellites (IASI, CrIS, AIRS, TES) without far-IR counterparts, and the construction of far-IR simulated radiances consistent with the mid-IR observed radiances could be very useful'. It would be a Herculean task to simulate every individual spectrum, and in many cases sufficient knowledge of the underlying conditions would not be available to do so, so deriving a robust statistical approach to perform the task is attractive. Ultimately one would want to both (a) validate our understanding of the far-IR and (b) test any such statistical approach with suitable observations, such as those proposed via the FORUM mission.

Our point in showing the noise-free LBLRTM results (applied to the observations) is to indicate how instrument noise affects both the choice of predictor wavenumbers from a noise-free case, at least in 'model' world.

Referee #3 - 17: Specific comments by page, line. More important comments denoted by *: Section 1 2, 3 – perhaps use "as much of as N %" instead of "significant" since "significant" is pretty subjective (and the fraction is probably less than 50%)

Response 17: Changed for "allow as much as 45 % of FIR radiation emitted from the ground to escape directly to space" (from Harries et al., 2008).

Referee #3 - 18: 2,9 – some extra "v" characters are present

Response 18: This has been corrected.

Referee #3 - 19: 2, 13 – It might be worth mentioning the RHUBC campaign (papers by Turner, Mlawer)

Response 19: This has been corrected.

Refere #3 - 20: 3,1-3 – A potential satellite mission is not relevant to this paper's purpose since this analysis only applies to ground-based measurements. It may be relevant to the T15 paper, but not to this paper.

Response 20: See answer to point 14, 24-29.

Referee #3 - 21: Section 2 3,15 – The REFIR_PAD goes up to only 1400 cm-1, so it's hard to see how that could be a "complete longwave dataset."

Response 21: "Complete" has been removed.

Referee #3 - 22: 4,9 – Perhaps start this sentence "Each radiosonde records data every 2 seconds: : :". The current wording might be taken as sondes are launched every 2 seconds.

Response 22: We could not find the sentence "each radiosonde records [. . .]" but the closest sentence has been rewritten for "During a radiosonde launch, data are recorded every 2 seconds, corresponding to around 800 measurements [. . .]"

Referee #3 - 23: 4,19 – Perhaps get rid of "developed by" and change Clough et al.

into a regular reference (i.e. in parentheses). The current wording makes it seem like the model was developed around 2006.

Response 23: This has been corrected.

Referee #3 - 24: 5,1 – In the far-IR, the linefile has substantial modifications to the HITRAN 2016 widths following Delamere et al. and Mlawer et al. The latter study also led to MT_CKD_3.0. The text says "includes modifications", but doesn't specify what the modifications were in reference to (i.e. modifications to what?). Since the RHUBC-II results are primarily based on REFIR-PAD measurements, it is probably worth mentioning when introducing the model.

Response 24: Following http://rtweb.aer.com/lblrtm_whats_new.html , I rewrote as: "[…] code MT_CKD_3.0 which includes up-to-date H2O foreign continuum from 0-600 cm-1 and the self continuum in the microwave that resulted from an analysis of measurements taken at the ARM RHUBC-II campaign and a re-analysis of RHUBC-I measurements."

Referee #3 - 25: 5,22 – Do the conditions have to be clear for the entire period between when the sonde is launched through the REFIR-PAD measurement time?

Response 25: From Rizzi et al., 2016, the approach to separate clear sky to cloudy sky uses radiances ratio as specific wavenumbers. Therefore, it implies that the cloud condition does not change much during the measurement time. Daily lidar quick looks have also been manually used to verify the classification. In addition, since the acquisition time is 6.5 min, it seems safe to assume that the weather conditions are stable.

Referee #3 - 26: 5,25 – random subsampling?

Response 26: Yes, the subsampling was performed with running random indexes. A few sets of random indexes has been tested to check if the subsampling had an impact and it did not.

Referee #3 - 27: Section 3 6,8 and Fig. 1 – Specify whether these results are for linear

or log.

Response 27: The correlation map is built by looking at the correlation between a vector based on the radiances at a specific FIR wavenumber and the corresponding vector based on the radiances at a specific MIR wavenumber. Therefore, there is no linear or log consideration at this stage. Same for figure 1 where the example originates from the data and not an extension. The linear and log consideration applies for figures 3, 4 and 6. In these cases, the text specifies linear or log. But this has also been added to the figure captions.

Referee #3 - 28: *9,2 – The noise added is between -1 and 1 times the standard deviation of the measured noise at each frequency. Doesn't this underestimate the actual noise? Why not use the chosen random number to sample from a normal distribution with that standard deviation?

Response 28: I am not sure that I understood properly your suggestion since a normal distribution with a standard deviation of $\sigma$ is the same of a normal distribution with a standard deviation of 1 times the standard deviation $\sigma$ (our case): np.random.normal(mu, sigma, 10000) $\equiv$ np.random.normal(mu,1.0, 10000)*sigma

Referee #3 - 29: Also, does the actual noise have spectral correlation? (i.e. if a case is higher than the mean in the MIR, is it likely to be higher than the mean in the FIR?). If so, then not taking that into account in the method may lead to inferior results for the LBLRTM+noise compared to using a pure measurement approach.

Response 29: The error of REFIR-PAD measurements were considered completely uncorrelated. This is very common in most of previous analysis of this kind of measures from other instruments similar to REFIR-PAD, see for example (Turner, 2005). In this study we considered the standard deviations of the mean of 4 spectra (or equivalently the NESR, which is similar). The calibration error, which actually could show some correlations over the frequencies, it is negligible with respect to the main statistical component. Ref: Turner, D. D.: Arctic Mixed-Phase Cloud Properties from AERI Lidar

Observations: Algorithm and Results from SHEBA, J. Appl. Meteorol., 44, 427–444, 2005.

Referee #3 - 30: 9, 25-29 – It should be made more clear that all these LBLRTM-based regression approaches are being applied to the MIR observations and not the LBLRTM simulated radiances.

Response 30: This has been clarified.

Referee #3 - 31: 10,11 – Using the mean difference may allow some cancellation of errors between the spectral points that are being averaged. It might be better to use the mean absolute value of the differences.

Response 31: The absolute mean value of the differences is not much different from the mean difference since biases between extensions and the original observed spectra tend to follow trends in some spectral bands (positive between 300 and 550 cm-1 and negative between 170 and 300 cm-1) as seen in Figure 4 and 6.

Referee #3 - 32: Section 4 10,15 – "exhibited" would be better than "exemplified". Also, "affect", not "effect".

Response 32: These have been corrected.

Referee #3 - 33: *11, Fig 6 – I think that the result that is plotted is from a single case. If so, please label it as such. However, if true, that opens up a more serious critique. Until the paper gets to Table 1, all results shown and discussed are from an example, not from the full dataset. How can the reader know whether these results are representative of the entire dataset?

Response 33:Results in figure 6 are not related to a single case, but using the entire set together (as well as in figure 4b). Only figures 1, 3 and 4a are related to a specific case as examples.

Referee #3 - 34: *12,9-15 – This section is puzzling. It is not up to a user's discretion

whether to interpret the relative humidity measured by a sonde as being with respect to liquid or ice saturation pressure. This is determined by the sonde design and processing software, and is done with respect to liquid. Interpreting it with respect to ice is not correct. In addition, it is difficult to understand the logic behind the statement in lines 13-15 "(indirectly implies : : :"). Why would changing the poor results obtained from sonde water vapor profiles obtained by the method described in this paragraph have anything to do with applying the methods in this paper to other conditions? This entire paragraph should be deleted.

Response 34: Actually, this paragraph was deleted from a previous draft of the article which was not submitted. As seen online (and on the ACP editor board), the actual submitted paragraph at page 12 between lines 9 to 15 is related to studies on the impact of the chimney.

Referee #3 - 35: 12,21 – Perhaps add a few words to clarify: ": : : the vertical resolution and assumptions made in our modelling approach without adding a chimney layer are sufficient : : :"

Response 35: We performed tests on the difference between observations and simulated spectra assuming the first level being outside the chimney and the first level being inside the chimney. The difference between both cases appears to be lower than 0.1 r.u. (below 600 cm-1) and peaking at 0.2 r.u. in the centre of the $CO_2$ band (as seen on figure r6). This information has been added to the text.

Figure r6: Blue: REFIR-PAD – LBLRTM using a first level from the chimney mean temperature from 2016. Red: REFIR-PAD – LBLRTM with our basic vertical resolution. The red curve is superposed below the blue curve. Green: LBLRTM with our basic vertical resolution – LBLRTM with the test first level.

Referee #3 - 36: Section 5 *14,1 – The Rizzi et al. paper certainly shows that current spectroscopy is sufficient to match observations and is an improvement over previous spectroscopy. The results in this paper show nothing of the kind. Perhaps the LBLRTM

results the authors performed also indicate this. However, these results have not been presented in this manuscript.

Response 36: Indeed, the writing of the corresponding sentence tends to mislead the reader as we did not present in the article comparisons between REFIR-PAD data and the corresponding simulations (as seen in figure r7). The part of the sentence has been removed (taking into account the next point too).

Figure r7: Mean radiance differences between REFIR-PAD and LBLRTM for the whole year (black), summer time (red) and winter time (blue).

Referee #3 - 37:*14,2-3 – The paper has not shown that an unbiased atmospheric state is essential to the approach that has been demonstrated. Water vapor profiles from sondes under very dry conditions are known to be biased (e.g. Miloshevich et al. (2009)), so the profiles that are used in this paper are likely far from unbiased.

Response 37: The relative humidity measurements have been corrected by the algorithms of Miloshevich et al. (2009) as explained in Tomasi et al. (2011): "[...] For this purpose, use was made of (1) the lag error corrections estimated by Rowe et al. [2008] (R08 hereinafter) in the presence of strong temperature inversions near the ground at Dome C; (2) the algorithm of Cady-Pereira et al. [2008] (C08 hereinafter) for correcting the RS80-A solar heating dry biases affecting the relative humidity (RH) measurements, properly normalized to the dry bias estimates made by R08 at Dome C; and (3) the algorithms of Miloshevich et al. [2009] (M09 hereinafter) for correcting the RS92 instrumental errors affecting RH. Using these algorithms, more reliable evaluations of RH have been obtained than those achieved by T06 in April–May 2005. [...]" In addition, although we did not show it in the article, we changed and tested various RH profiles to assess the extension, with large biases seen.

Referee #3 - 38: *14,13-15 – As in the comment above, this really has not been shown. At best, the analysis about the water vapor saturation pressure over ice that is alluded to (but not really shown) suggests that this might be true, but this is far from being

demonstrated.

Response 38: This part was also in a previous draft of the manuscript and was not discussed in the submitted version.

Referee #3 - 39: 14,24-29 – It is unclear what the results from this ground-based study have to do with the possible future FORUM mission. The authors should make their argument here more clearly or abandon it.

Response 39: The possible future FORUM mission aims to provide the first FIR global data set observed from space (along with MIR data). T15 suggests that is it possible to extend MIR to FIR using a predictor model. However, this model has been built using LBLRTM simulations and has only been compared to broadband observations. Using the REFIR-PAD observations, we observe that such an approach can achieve acceptable accuracy if instrument noise is taken into account. Therefore, it seems feasible that a model to extend the MIR to the FIR as described by T15 could be built using FORUM data. This model would need to be assessed by the FORUM measurements as we have done here with REFIR-PAD data. If successful, such a model could be used retrospectively on IASI (after collocated verifications between IASI extensions and FORUM) to estimate long trends of FIR radiances for use in, say, model evaluation. Note that of course we are not saying that the regression model we have built here is suitable for this purpose, simply that the technique appears viable.

———————————————

**Fig. 1.** FIR radiances at 447.20 cm-1 against predictor MIR radiance at 1376.00 cm-1 for simulated LBLRTM cases (same as figure 3).

[Figure]

**Fig. 2.** Differences between downwelling radiances computed using an atmospheric profile with and without water vapour (in red), and for 2 cases of temperature perturbations (+/- 1 standard deviation, respecti

[Figure]

**Fig. 3.** Differences between downwelling radiances with local perturbations in the water vapour profile. In red, the VMR below 5 km has been increased of 10 %, in green (blue), the VMR between 5 and 10 km (bet

[Figure]

**Fig. 4.** Relative difference for the same cases as figure r2, between 650 and 800 cm-1.

[Figure]

**Fig. 5.** Relative difference for the same cases as figure r3, between 650 and 800 cm-1.

Fig. 6. Blue: REFIR-PAD – LBLRTM using a first level from the chimney mean temperature from 2016. Red: REFIR-PAD – LBLRTM with our basic vertical resolution. The red curve is superposed below the blue curve.

[Figure]

**Fig. 7.** Figure r7: Mean radiance differences between REFIR-PAD and LBLRTM for the whole year (black), summer time (red) and winter time (blue).

---

## Author Response (AR2)

Report #1 Submitted on 11 Feb 2019 Anonymous Referee #1

This paper is well written and presented. The problem is its very limited applicability (uplooking Antarctic atmospheric paths) but given the technique described here would be more widely applicable I have recommended it is published.

I noted on page 1 line 15 that the sentence needs to be changed to ... is accounted for in the simulation procedure.

 $\checkmark$

Report #2 Submitted on 13 Feb 2019 Anonymous Referee #2

The authors have responded in detail to the reviewers' comments. Notwithstanding a few remaining issues (see below) I think the paper is in good shape to be published. Detailed points:

1. It is welcome that the authors have clarified that their technique is only strictly tested on the REFIR-PAD data set in question, and would require validation for other climate regimes and/or viewing geometries.

2. Thank you for Fig. r1 showing predictor behaviour for the pure LbL case. One thing that's clearer to me on re-reading is the point made on page 11 line 15 that skewing the predictor channels to around 1400 cm-1 means using REFIR-PAD observations that have relatively high noise in Fig. 1. In other words, it's a bad idea to use noisy MIR channels to predict FIR channels.

3. Another reviewer commented that channels in the CO2 band may be predominantly sensitive to near-surface temperature. Figs. r2-r5 show some limited support for the idea that water lines also play a role. Another point worth considering is that for a satellite sensor the Jacobians will look very different to a ground-based sensor, e.g. there will be more sensitivity to the upper atmosphere cf. troposphere.

4. In their responses the authors say it "would be a Herculean task to simulate every individual spectrum" in the historic satellite MIR record as a means of computing the FIR contribution directly. I agree. (At least, the task of processing the required profile data from e.g. reanalyses would be very time consuming.)

5. I'm still confused by the terminology used for uncertainty. On page 7 line 14 the phrase "mean absolute standard deviation" is used. I simply don't know what this means since a standard deviation is always positive. Assuming the authors are talking about Fig. 4b perhaps

they are referring to the grey shaded area – in which case it is a standard deviation of a mean difference at 10 cm-1 resolution.

> In this exact part, the "mean absolute standard deviation" was the wrong term. We were speaking of the absolute difference mean and this has been changed accordingly.

6. Page 11, lines 22-24: "model... is capable of capturing the [radiance] to within 2%, except in selected bands... with a peak at 540 cm-1, see figures 6(a) and (b)". I think "peak" is confusing here, perhaps "largest discrepancy" or similar which doesn't have connotations of a positive difference.

> This has been changed for "the largest discrepancy".

7. Page 13, line 15: "(up to -119%)", do to the authors mean -124% as in the table entry? ☑

Report #3

Submitted on 07 Mar 2019 Anonymous Referee #3

Based on the replies to reviewers and the revised version, here's how I see the main questions on which a publication decision should be based:

Is there an application to which the approach demonstrated in this paper can be applied? The answer is arguably 'yes'.

Let's say FORUM has been launched, and a study is produced that shows that its far-IR radiances can be predicted reasonably accurately by FORUM radiances measured in other regions. That demonstration will suggest that a similar approach would work using the large historical record of IR observations (e.g. AIRS, IASI), thereby allowing IR measurements over decades to be "extended" to the potent far-IR. Therefore, if this paper had been able to use FORUM measurements instead of ground-based REFIR-PAD measurements, then the answer would clearly be 'yes'. So the argument for the paper under review being publishable hinges on whether the method demonstrated for the REFIR-PAD extension from the IR to FIR is germane to a similar possible future extension for satellite instruments.

Note that the application of the method in this paper to other ground-based instruments is not worthy of publication; the authors seem to agree with this point. There are only a couple of ground-based spectral IR instruments (i.e. does not measure in the far-IR, e.g. AERI) deployed in locations in which the far-IR is not always opaque, so there would be minimal need for a method to extend these data records.

How similar are the ground-based and satellite-based situations?

To see how germane the REFIR-PAD extension presented in this paper would be to an extension for a satellite-based instrument, the similarities and differences between the two

different viewing geometries with respect to the relationship between far-IR and IR radiances must be explored.

- What would the satellite see? In the far-IR, radiances would depend on the water vapor and temperature profiles – a very rough rule of thumb is the observed brightness temperature in an instrument channel would be the temperature at the height at which the integrated optical depth (primarily due to water vapor in this spectral region) from the top of the atmosphere to that height is about 1. There would clearly be similar channels in the nu2 band of water vapor, so there is every reason to expect that a good extension to the far-IR from the nu2 band could be developed. The nu2 band clearly has all information about the tropospheric water vapor field (since it is used for water vapor retrievals), and could also presumably be used for temperature retrievals. There are two important caveats to this. First, since the far-IR water vapor band is stronger than the nu2 band, the dependences on water vapor concentrations higher up (e.g. stratosphere) than the vertical region that the nu2 band is sensitive to might impose some limitations to the extension. Second, radiances in spectral channels in carbon dioxide bands would have little correlation with the far-IR radiances and wouldn't be used in a single-channel satellite-based extension. (This paper considers only single-channel extensions.)

- What does the ground-based REFIR-PAD see in a location like Antarctica? Qualitatively, there are three types of far-IR channels (x-axis on Figure 2):

1) Purely opaque channels – These are everywhere < 200 cm-1 and where there are strong water vapor lines throughout the rest of the far-IR. Radiances in these channels are sensitive only to the temperature very near the instrument. These cases are not similar to any satellite-based channels.

2) Mostly opaque channels – Microwindow regions from 200-400 cm-1 and near some relatively strong lines from 400-600 cm-1. Radiances are sensitive to the temperature and water vapor profiles, in particular those values closer to the surface. This category is somewhat similar to satellite-based channels, although the vertical range of the profile that matters is probably somewhat smaller than for corresponding satellite channels.

3) Semi-transparent channels – Everywhere in the 400-600 cm-1 that is not near a relatively strong line. Radiances are sensitive to the water vapor and temperature profiles, with the sensitivity ranging higher than in the category above (possibly including the water vapor column). This category is pretty similar to corresponding satellite channels.

The answer to the question in this section ("How similar are the ground-based and satellite-based situations?") is "in theory, partly similar".

In actuality, how much of an analogue to a potential satellite extension is the ground-based extension presented in this paper?

Since the extension for the opaque channels have no dependence on water vapor, they have no analogue in the satellite case and, therefore, the results shown in the paper are not germane to the satellite case. (For the surface case, any opaque channel, whether co2- or h2o-dominated, will be able to predict the radiance.) The 'mostly opaque' channels do have analogues in the satellite case -- there are regions of the y-axis of Figure 2 that have similar optical depth dependences (e.g. in 1300-1400 cm-1), which would be the case for both the ground-based and satellite-based perspectives. So channels in this category have the potential to be good analogues for the satellite case, and are potentially germane to the main question. However, the high noise of the instrument from 1300-1400 cm-1 makes this region not sufficiently predictive for the corresponding far-IR points, which therefore get "matched" with spectral points in the CO2 band, i.e. with virtually no sensitivity to water vapor. This results in reasonable accuracy for the extension, but that is irrelevant to the question of whether this result is germane to the more important question at hand. Using a temperature channel from a satellite instrument to predict far-IR radiances from that instrument clearly wouldn't have the proper sensitivity, so the results from this category are clearly not germane. The third category (semi-transparent) is most similar to the 1300-1400 cm-1 and 760-800 cm-1 regions on the y-axis. Again, the 1300-1400 cm-1 does not work well due to noise, so the best match is indeed from spectral channels in a region with similar dependences on water vapor and temperature. If this were a satellite-based exercise, I would expect these same far-IR channels to also be fairly well modeled by the same 760-800 cm-1 channels.

The answer to the question posed in this section is "a very limited analogue".

Is the study germane enough to the satellite case to justify publication?

The entire argument that the extension to the far-IR shown in this paper is germane to the satellite case rests on the somewhat limited number of channels from ~500-620 cm-1 that are 'matched' with IR channels from 760-800 cm-1. That is a very limited result.

The answer is 'no'.

For the sake of argument, assuming that this result is sufficiently germane, is the methodology presented something that someone might consider using for a satellite-based extension to the far-IR?

This study determines a single IR channel to match each FIR channel. The arguments presented above about which region has channels that best match categories of FIR channels are very qualitative. In actuality, a channel in one region will not perfectly match the dependences on water vapor and temperature in a different region. In Huang et al., a multivariate fit of IR measurements is used to simulate the far-IR region. In the generalized training for OSS, a number of monochromatic calculations in different spectral regions are needed to match channel radiances. If one were developing an extension of (say) IASI to the far-IR, it would be limiting and foolish to use a single channel.

The answer to this question is 'no'.

**Summary**

My perspective on this paper has not changed since the first review, in which I wrote that this "paper suffers from significant motivational and methodological issues." I do not think it should be published.

> Firstly, we would like to thank the reviewer for the time they have clearly spent reading the manuscript. We do not disagree with many of the insightful points they make regarding the spectroscopy and have been at pains to point out in the revised manuscript that we are aware of the limitations of the study with regards to the specific viewing geometry and conditions sampled, something acknowledged by the other reviewers.

The main finding of our work is the need to take careful account of the instrumental characteristics (especially noise) if one is creating synthetic far-infrared spectra from real mid-infrared observations. This may not be surprising but does not appear to have been explicitly considered in publications that have tried to do exactly this in the past. Essentially, we show that noise can seriously limit the usefulness of channels that one would expect to contribute the most information in reconstructing the spectral behaviour of semi-transparent regions in the far-infrared. For this particular instrument and viewing geometry, the situation can be partially rectified by exploiting information in the wing of the 15 micron  $CO_2$  band but we make it clear in our discussion that this may not hold for a satellite instrument viewing in nadir. Our results explicitly show that for any method seeking to exploit future far-infrared satellite based measurements to synthetically extend earlier mid-infrared records the noise behaviour of both instruments will need to be properly taken into account.

The main remaining objection to publication appears to be that the method applied here (using single channels as predictors) has limited applicability to satellite based measurements. This is despite the fact that the same underlying method has, in fact already been applied to IASI data at the global scale, with the results published in this journal (as noted in the manuscript).

**Can downwelling far-infrared radiances over Antarctica be estimated from mid-infrared information?**

Christophe Bellisario1, Helen E. Brindley2, Simon F.B. Tett1, Rolando Rizzi3, Gianluca Di Natale4, Luca Palchetti4, and Giovanni Bianchini4

[revised manuscript text omitted]